# Context-Aware Personalization: A Systems Engineering Framework

**Olurotimi Oguntola** * and **Steven Simske**

Systems Engineering Department, Colorado State University, Fort Collins, CO 80523, USA;
steve.simske@colostate.edu
* Correspondence: timi.oguntola@colostate.edu

**Abstract:** This study proposes a framework for a systems engineering-based approach to context-aware personalization, which is applied to e-commerce through the understanding and modeling of user behavior from their interactions with sales channels and media. The framework is practical and built on systems engineering principles. It combines three conceptual components to produce signals that provide content relevant to the users based on their behavior, thus enhancing their experience. These components are the 'recognition and knowledge' of the users and their behavior (persona); the awareness of users' current contexts; and the comprehension of their situation and projection of their future status (intent prediction). The persona generator is implemented by leveraging an unsupervised machine learning algorithm to assign users into cohorts and learn cohort behavior while preserving their privacy in an ethical framework. The component of the users' current context is fulfilled as a microservice that adopts novel e-commerce data interpretations. The best result of 97.3% accuracy for the intent prediction component was obtained by tokenizing categorical features with a pre-trained BERT (bidirectional encoder representations from transformers) model and passing these, as the contextual embedding input, to an LSTM (long short-term memory) neural network. Paired cohort-directed prescriptive action is generated from learned behavior as a recommended alternative to users' shopping steps. The practical implementation of this e-commerce personalization framework is demonstrated in this study through the empirical evaluation of experimental results.

**Keywords:** context awareness; intent prediction; persona; e-commerce personalization; systems engineering



## 1. Introduction

The industrial experiences of both practitioners and researchers demonstrate the growing importance of technology-enabled omnichannel customer experiences in retail. Successful retail companies are paying attention to changing shopping habits and the increasing inclination to shop digital first, as they record continually more store purchases that started with online product searches. The personalization of an e-commerce website gives it a competitive advantage by simplifying users' decision processes using relevant information. There have been research efforts to model adaptive systems, implement personalization, and utilize context in recommender systems. They have been applied to many areas, such as e-commerce, tourism, entertainment, and social media [1–6]. These papers delve into different machine learning methods to apply context to generate personalized recommendations for various applications. However, there is a shortage of exhaustive theoretical and practical coverage of the actual implementation of these systems in practice. Industry practitioners often find a disconnection between research outcomes and their applicability in practice.

Interestingly, the personalization process, if generalized, may not have the desired impact: effective personalization is perceived to be environment- and domain-specific [7,8].

A systems engineering framework is apt for the complexities of the data foundation requirements, decision-making, and integration between the multiple knowledge-generating subsystems required to provide intuitively personalized shopping experiences to the increasing traffic on shopping channels [9].

This study highlights the development and implementation of a systems engineering-supported framework that is employed to enable the realization of a system that provides positive and productive shopping experiences that mimic what a user will get from the attention of a knowledgeable shop assistant. The "systems engineering framework" uses a hierarchical approach from traditional systems engineering, with top-down decomposition and the definition of development requirements for its subsystems, along with bottom-up integration and qualification to validate against specified requirements. The latter include tests, measurements, and reliability and sensitivity recommendations.

The framework, which is inspired by the systems engineering approach of "faster, cheaper, better", which emphasizes the use of Commercial-Off-The-Shelf (COTS) systems and components [10], is intentionally designed to be simple, scalable, and replicable with the open-source technology that is easily accessible. The proposed system, which fits to loosely coupled microservice-based e-commerce platforms, produces signals that activate products and non-product responses that are personalized to individual users in their current shopping session. This is implemented through the following specialized subsystems:

- Persona—'recognition and knowledge' of the users and their behavior;
- Awareness of users' current context;
- Intent prediction—comprehension of their situation and projection of future status;
- Cohort-directed prescriptions for the next best content.

Logically, a system with this framework enabled qualifies as a microservice [11] since it can be deployed, scaled, tested independently, and be responsible for producing signals for hyper-personalized in-session shopping experiences. Similarly, the persona generation, context-aware computing, and in-session intent detection subsystems qualify as microservices. These subsystems are the building blocks that meet requirements and applicable system design criteria. The persona generation subsystem satisfies the 'knowledge and recognition' system requirement by grouping users into cohorts and aggregating online behavior for user cohorts to build behavioral features. The context-aware computing subsystem's analytic computations provide the users' current contexts. Finally, the in-session intent detection subsystem uses the outputs of the other subsystems to accurately predict the intention at every stage of the shopping journey to generate relevant signals and contents, thus providing a personalized user experience. The intent detection is extended by a novel method that implements cohort-directed alternative action prescriptions. To the best of the authors' knowledge, no other study before this has proposed a systems engineering framework for an end-to-end practical and scalable implementation of personalization that leverages privacy-compliant customer profiling methods, adopts current context, and influences users' in-session actions with cohort-directed prescriptions for the next best content for delivery.

The rest of the paper is organized as follows. Section 2 describes the system design considerations. It explains the subsystems and their inputs and outputs, including the requirements at the system level, the system synthesis, its analysis, and the implementation of a personalization task. We present the results in Section 3. In Section 4, we evaluate and discuss the results. Section 5 concludes the paper and outlines its limitations and important future work.

## 2. Materials and Methods

### 2.1. Data Capture

The first key input into the personalization system is the users' first-party behavioral data. The data used for proof of concept are publicly available e-commerce behavior data [12], and the data tracking used to produce embeddings is smaller than for standard

e-commerce use cases such as re-targeting [13]. This implementation requires effective and accurate capturing, storage, and the easy accessibility of first-party data on users' interactions on shopping channels, including clicks, swipes, searches, product views, page views, cart additions, orders, returns, reviews, etc. Website visitor information is mined from log files and page tagging through web cookies, and many tag management solutions have been developed to do this effectively. However, cookies have heightened internet privacy concerns, despite their usefulness for web data collection, because third parties can exploit them to track user behaviors. To address these privacy concerns, privacy-preserving data mining and data publishing methods are used to incorporate the preferences and requirements of the stakeholders [14,15].

With awareness of the cookie-less future, this personalization system leverages first-party data, including data from user registration, user submissions, and behavioral data collected through privacy-compliant device fingerprinting mechanisms [16]. The user-agent header, the browser used, the approximate location based on IP address, and screen resolution are a few individually non-identified attributes that, when combined, hold valuable identification properties used for privacy-compliant behavioral tracking online. Figure 1 is a high-level schematic diagram of the architecture of the personalization system. Platform tag management is depicted in the Data Capture Layer in Figure 1 but is external to the personalization system. Its implementation is beyond the scope of this experimental design.

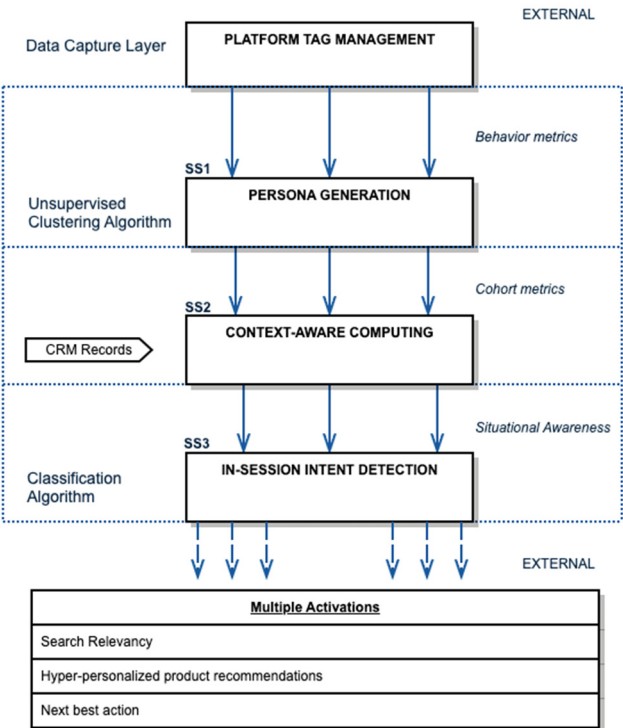

**Figure 1.** Personalization System Architecture.

This framework is considered data-centric since data are its primary asset and the dominant system component [17], and feature engineering [18], of the raw data collected plays a significant role in its implementation. With the maturation of machine learning approaches and because data quality is so essential [19,20], the requirements for good quality input data are emphasized in implementing this system through feature engineering and data-centric iterations to obtain improved performance from the models. Details of the data preprocessing and transformations employed in the implementation of this research are covered in the subsystems.

*2.2. Persona Generation*

The persona generator is the first subsystem in the proposed personalization systems engineering framework. It groups users with similar behavioral metrics into cohorts. The feature engineering process that selects and transforms the most relevant variables from the raw data input leverages retail domain knowledge. It requires an understanding of the underlying features in the raw data that best describe the events, and knowledge of how to transform them into the format of input needed. As well as being optimal, the choices of processes and algorithms within this framework are made intentionally for replicability.

The persona generator initiates the conversion of vast amounts of data on user behavior into meaningful insights. This subsystem's user groups or cohorts are created without knowledge of the desired outcome; therefore, the input data are not labeled. In other words, the data samples are not tagged with one or more labels that provide context for the machine learning model to learn from, as is done in traditional supervised learning. As it is in many practical applications, collecting labeled data is expensive and laborious, and as a result workaround solutions developed to handle unlabeled and sparsely labeled data [19,20] were explored and adopted for this implementation. Unlike a classifier, the persona generator has no requirements for matches to specific outcomes. Clustering, an unsupervised machine learning technique, identifies and groups similar data points into structures and is optimal for this subsystem. Many clustering techniques use the degree of similarity between records for their grouping [21]. The clustering process produces a set of distinct groups but with the objects within each group being broadly similar according to the relevant metrics. The clustering techniques considered for this component include hierarchical clustering, DBSCAN (density-based spatial clustering of applications with noise), K-Means, and Gaussian mixture models. Different popular clustering algorithms include BIRCH (Balanced Iterative Reducing and Clustering using Hierarchies), Agglomerative Clustering, Affinity Propagation, Mini-Batch K-Means, Mean Shift, Spectral Clustering, and OPTICS (Ordering Points To Identify Cluster Structure). Hierarchical clustering is suitable for finding a hierarchy of clusters called a dendrogram, which, as the name implies, is structured like a tree [22,23]. DBSCAN is a technique apt for discovering clusters of different shapes and sizes in noisy data and is particularly useful for spatial data mining.

The K-Means clustering algorithm is well-known for its simplicity and ease of deployment in practice. The technique randomly initiates K points as the initial centroid of the K number of clusters in its basic form. Then, it iteratively assigns data points to one of the K clusters based on its Euclidean distance to the cluster centroid in the feature space. The cluster centroid is determined in each iteration by computing the mean of all the data points belonging to that cluster, and the whole process is repeated until convergence. If we denote the set of features as $X = (x_1, x_2, \ldots., x_N)$, then the K-Means process [24], as described, can be expressed in mathematical notation as:

i.　Random initialization of cluster centroids $\{z_1, z_2, \ldots, z_k\} \in \mathrm{R}^N$

ii.　Iterate until convergence:

$$
\begin{cases}
\text{For each i, set} \\
\quad e^{(i)} = \arg\min_j \left|\left| x^{(i)} - z_j \right|\right|^2 \\
\text{For each j, set} \\
\quad z_j = \dfrac{\sum_{i=1}^{m}\left\{ e^{(i)}=j \right\} x^{(i)}}{\sum_{i=1}^{m}\left\{ e^{(i)}=j \right\}}
\end{cases}
\tag{1}
$$

It is typical to test values within a reasonable range to determine an optimal value for K. One method is to select the value that gives the highest ratio of between-centroid variability to within-cluster variability and leverage regularization methods in an optimization process to determine K [25]. Different solutions have been developed to address the K-Means algorithm's susceptibility to initial cluster centers and its vulnerability to outliers and noisy

data [26,27], as well as its adaptability to different data types, including time-series and multi-view data [28,29].

Instead of simply associating each data point with one and only one cluster, Gaussian mixture models introduce a probability that indicates how much a data point is associated with a specific cluster. Fundamentally, a Gaussian mixture is a function of several Gaussians. When applying the Gaussian mixture model [30] to create K number of clusters from a set of observed features $X = [x_1, x_2, \ldots, x_N]^T$, each Gaussian has a mean $\mu \in R^N$, a covariance $\sum \in S^N$, and a mixing probability $\pi$, and its probability density function is given by

$$p(x; \mu, \textstyle\sum) = ((2\pi)^{N/2} \mid \textstyle\sum \mid^{1/2})^{-1} \exp(-^{1/2}(x - \mu)^T \Sigma^{-1}(x - \mu)) \tag{2}$$

The convergence of the Gaussian mixture model is based on the expectation-maximization algorithm [31], and the process can be summarized as follows:

i. First, initialize mean $\mu$, covariance $\sum$, and mixing probability $\pi$;
ii. Evaluate the initial value of the log-likelihood L;
iii. Evaluate the responsibility function using current parameters;
iv. Using newly obtained responsibilities, obtain the new $\mu$, $\sum$, and $\pi$;
v. Compute the log-likelihood L again. Iterate (iii) and (iv) until convergence.

The first subsystem, the persona generator, builds user cohorts from the users' shopping activities on the shopping channel. Typical clickstream data has records of users' click behavior, details of products interacted with, and events, one of which is usually the desired outcome of conversion (cart checkout). These form the user's click behavior sequence with variables $\{x_{UID}, x_{SID}, x_{P1}, x_{P2}, \ldots, x_{PN}, x_{EV}\}$, which are user identity $x_{UID}$, user session identifier $x_{SID}$, N variables related to the products a user interacts with (both categorical and numerical) xPI, and the event type xEV. Leveraging domain knowledge, we transformed these signals into features of behavior metrics aggregated at the user level.

The features described in Table 1 were extracted from an e-commerce dataset collected by the Open CDP project from a multi-category store containing the online behavior data of 285 million consumers over several months [12]. The dataset includes sessions of timestamped events of users' product views, adding to cart, and purchases. The dataset also provides product ID, product category, category ID, brand, and price. The subset of the data used for synthesizing the persona generator subsystem is the first month's users' shopping activities; that is, 22.5 million timestamped events.

The system design optimization techniques applied in data preprocessing include standardization and principal component analysis (PCA). Because features have significant differences between their ranges, they are standardized by subtracting the mean and scaling to unit variance. To extract strong patterns from the large dataset, which has multiple features, we passed it through PCA and reduced the dimension to four principal components. The selection of four components was determined from the number of dimensions for which the cumulative explained variance [32] exceeded a threshold of 80%. Reducing to three and five principal components gave similar results. This preprocessing prepares the dataset for the clustering algorithms that group the users into cohorts based on their persona or patterns of interaction with the sales channel. After exploratory data analysis and considering multiple clustering approaches against the design requirements, we reduced the list of design alternatives to K-Means and Gaussian mixture models (GMMs) and further evaluated these unsupervised clustering methods. This approach follows the systems engineering practice of trade-off analysis, and the argument in [25] on the importance of building ground-truthing into system design at the onset by comparing and contrasting two or more frameworks at the system level.

**Table 1.** Persona generator features.

| Feature | Description |
| --- | --- |
| $\{\delta tEV = t_{i+1} - t_i\}$user_id | The time lapse between user events |
| $\{|X_i|\}$user_session | Count of events per user session |
| $\{\sum|S_i|\}$user_id | Cumulative count of sessions per user |
| $\{(\sum P_i/N_i)\}$user_id/purchase | Average order value per user |
| $\{|B_i|\}$user_id/purchase | Count of unique brands purchased per user |
| $\{\sum|X_i|\}$user_id/purchase | Number of purchase events per user |
| $\{|C_i|\}$user_id/purchase | Count of unique product categories purchased per user |
| $\{(\sum P_i/N_i)\}$ user_id/add_to_cart | Average price of cart per user |
| $\{|B_i|\}$user_id/add_to_cart | Count of unique brands added to cart per user |
| $\{\sum|X_i|\}$user_id/add_to_cart | Number of addition-to-cart events per user |
| $\{|C_i|\}$user_id/ add_to_cart | Count of unique product categories added to cart per user |
| $\{(\sum P_i/N_i)\}$user_id/views | Average price of products viewed per user |
| $\{|B_i|\}$user_id/views | Count of unique brands viewed per user |
| $\{\sum|X_i|\}$user_id/views | Number of product view events per user |
| $\{|C_i|\}$user_id/ views | Count of unique categories of products viewed per user |

To determine the optimal number of clusters for the input dataset, we iteratively tested with K ranging from 1 to 10 and adopted the Kneedle algorithm [33] to identify the 'knee' or 'elbow' point where the relative cost to increase the number of clusters is no longer worth the corresponding performance benefit. Figure 2 shows the line plot between SSE (Sum of Squared Errors) and the number of clusters; however, it does not depict the elbow point, which represents the tipping point where the SSE or inertia starts to decrease linearly. A low inertia and low number of clusters better meet requirements.

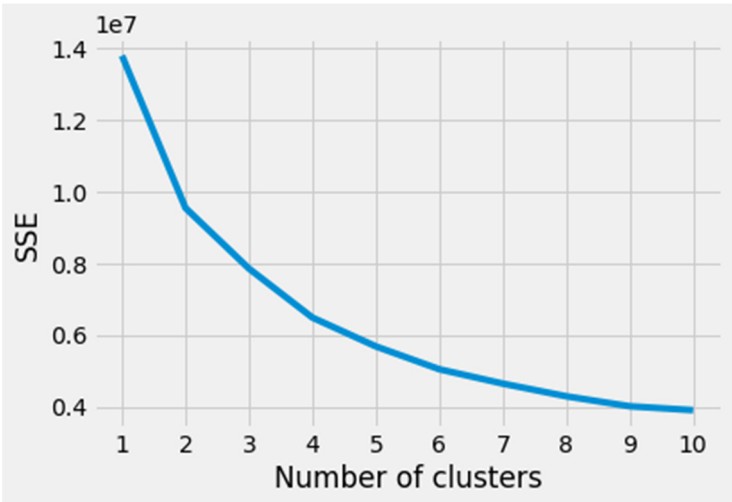

**Figure 2.** Sum of Squared Errors plot for K-Means clustering.

Figure 3 uses Yellowbrick's KElbowVisualizer [34] to more clearly visually identify the optimal number of clusters to be 5. The scoring parameter plotted computes the sum of squared distances from each point to its assigned center. The amount of time it takes to train the clustering model per K value is plotted in the intersecting dashed green line.

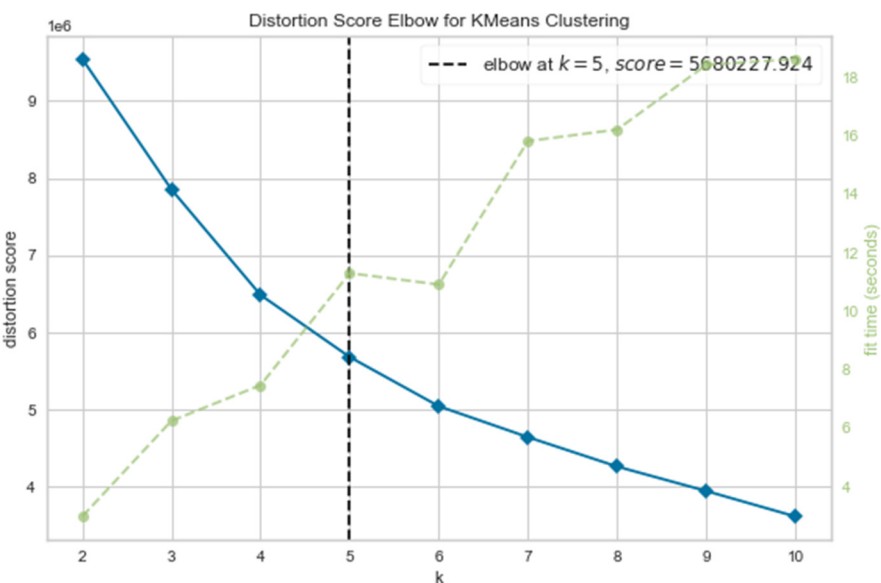

**Figure 3.** Distortion Score Elbow for K-Means clustering—optimal at K = 5.

Figure 4 shows the optimal number of clusters for the GMMs. The first prominent elbow of the AIC (Akaike information criterion) plot is at 5 clusters, which agrees with the optimal number of clusters identified in the K-Means model and is practical for this use case. After ascertaining the optimal number of clusters, we compare K-Means clustering with the GMMs for the exact optimal number of clusters. The first comparison method scores the clusters using the variance ratio criterion—the ratio between the within-cluster and between-cluster dispersion using the Calinski-Harabasz (CH) index [35].

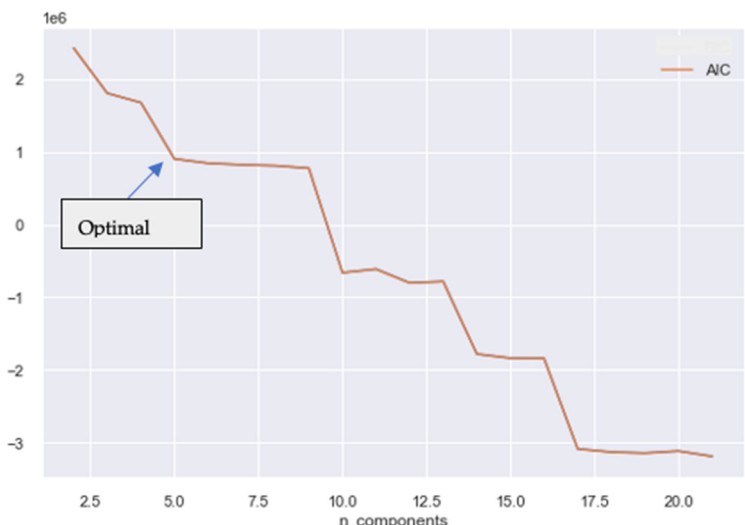

**Figure 4.** Optimal number of clusters for the Gaussian Mixture Models.

The K-means cluster had a higher CH index, indicating the clusters are dense and better separated. The second comparison method used is the Davies-Bouldin score [36], which measures the average similarity of each cluster with its most similar cluster. The K-means clusters had a lower Davies-Bouldin score than the GMM, indicating better clustering. K-Means was, therefore, adopted for clustering users into cohorts, which forms an input into other subsystems within this personalization framework. Extraction of the mean of a few metrics of users in the different persona cohorts formed indicates the separation between clusters (see metrics highlighted in Table 2). Cohort 0 has the most significant number of users but a relatively low number of product views per user, and even lower

purchases per user compared to Cohort 3, which has fewer users but much higher product views and purchases per user. The average time the users in these two cohorts take between user sessions is also very different.

**Table 2.** Metrics of the persona cohorts.

| Cohort | Size | Cohort Metrics | | | | | |
| | | Time between Sessions (s) | Events in Session | Sessions by Each User | Product Views | Add to Cart ($) | Purchases |
|---|---|---|---|---|---|---|---|
| 0 | 659,044 | 65,904 | 1.77 | 4.23 | 2757 | 2.43 | 0.06 |
| 1 | 119,238 | 27,981 | 7.67 | 31.52 | 16,159 | 15.34 | 0.33 |
| 2 | 105,160 | 45,791 | 3.86 | 17.94 | 10,036 | 802.4 | 1.49 |
| 3 | 6620 | 21,365 | 5.63 | 65.88 | 40,121 | 5928.21 | 13.75 |
| 4 | 14,866 | 9362 | 14.25 | 128.72 | 63,422 | 378.76 | 1.57 |

*2.3. Context-Aware Computing*

The second subsystem in the personalization systems engineering framework develops the context-awareness platform for acquiring contextual information and producing different results adapted to the context of use, further enabling the prediction of user action according to the user's current situation. Context-aware development within this framework complies with the variety of requirements that need to be satisfied to handle context. These include context interpretation, transparent and distributed communications, availability of context acquisition, separation of concerns, and context storage and history [37].

The functional architecture consists of subscriptions to suites of independently deployable services and an API gateway that accepts application programming interface (API) calls. The API gateway sends the calls through the API management system, collecting the various services for fulfilling the call and returning the appropriate result to the calling subsystem. The API gateway provides security, authentication, authorization, fault tolerance, load balancing, and routing [38]. Besides the many popular open-source API gateways that can be self-managed, major cloud service providers offer end-to-end API management in the cloud, on-premises, or as a hybrid.

The context development architecture grants us the flexibility to add or remove services that call for contextual information, depending on the scenario. For example, the architecture diagram below (Figure 5) depicts components for implementing context awareness in big-box retail. Microservices for such an implementation provide contextual information that includes the following:

A. inventory available in the user's location or preferred store;
B. user's product category and brand affinities;
C. order delivery location-based options;
D. proximity to stores and a selected store;
E. user's search queries;
F. directed response to inclement weather;
G. promotions available to a user;
H. availability of expert installation for certain products;
I. order history;
J. order tracking information;
K. the semantics of extracts from product reviews;
L. user's sensitivity to pricing;
M. customer lifetime value.

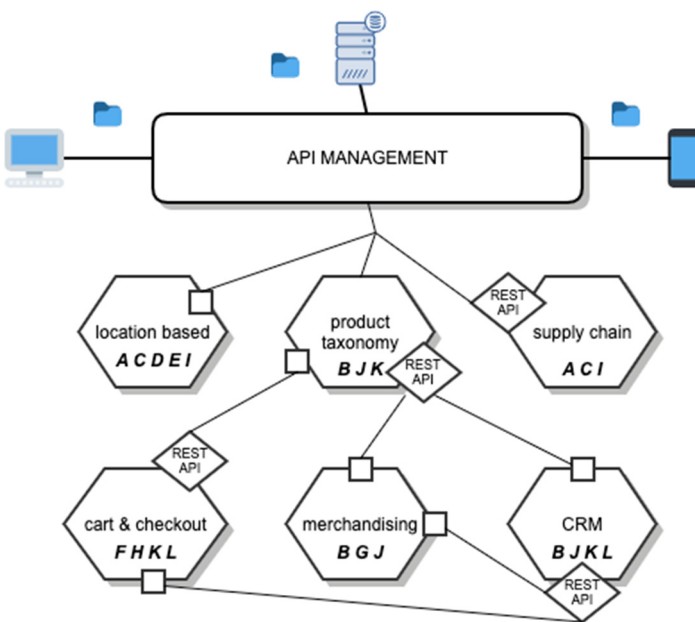

**Figure 5.** Context-awareness subsystem architecture.

Specific API calls to this subsystem will give a response based on the shopper's current context. For instance, a merchandising API call for promotions available to a user will provide promotion campaign codes for applicable price discounts to encourage purchase.

These microservices are built around business capabilities and exist as independently deployable enterprise applications. The architectural diagram depicts how multiple microservices provide the data combined for contextual information delivered to a browser, mobile device, or stored in a database. Microservices and API management technologies are research subjects of interest and have been adopted in the industry. API requests are integrated with business process models to provide documentation or the testing of REST API calls directly in a modeling environment [39]. Similar solutions are available as open-source technologies and can be deployed as a service. For the application in this paper, we pulled contextual information captured within the clickstream of interactions on the e-commerce website to validate the personalization of a systems engineering framework.

By design, the users' context variables are retrieved via API calls to microservices that provide different user context data in real-time or batched historical data stored in a low-latency data service application. Context variables will differ depending on the industry and requirements. Some variables that apply to this application include location-based, product taxonomy, supply chain, cart and checkout, merchandising, and customer relationship management microservices, as highlighted in the context-awareness subsystem architecture diagram in Figure 5. However, for this synthesis, the context variables (CV) listed below are computed from the input data:

1. $CV_1$—Propensity to purchase;
2. $CV_2$—Timelapse in the current session;
3. $CV_3$—Count of activities in the current session;
4. $CV_4$—Average price of products clicked through in the current session;
5. $CV_5$—Frequency of purchase;
6. $CV_6$—Measure of customer value ($CV_5 \times$ Average Order Value).

The propensity-to-purchase score is the probability of conversion given the independent variables and is computed by fitting the data pipeline to a logistic regression classifier predicting purchase. The rest of the context variables are as defined and are calculated for each user using their activities on the shopping channel over some time. Derived features that are powers and multiples of these were also experimented with (see Section 3).

### 2.4. User Intent Detection

The third subsystem in the framework is required to predict the user's intent in-session by deriving an understanding of the user's current situation and projecting the future status of the shopping session. The user's longer-term behavioral metrics, aggregated cohort metrics from the persona generator, and in-session contextual information, are all inputs into this module. The module predicts the intent of the user in-session, and for each step in the user's journey, if the prediction is contrary to the desired objective, computes signals leading to alternative action that better converges with the desired outcome.

Modeling users' interactions with customer engagement channels as sequences of actions is an intuitive method for the system to meet this requirement and has proven to be adequate in practical use cases [40,41]. For example, sequence models have been used for sentiment classification, machine translation, and music generation. In addition, they are effective for supervised learning in scenarios where either the model input or model output can be prepared as a sequence. Clickstream intent prediction is a challenging feature-based classification of sequences similar to language prediction [42]. The wide range of machine learning methods researchers explore to achieve clickstream intent prediction suggest applying deep learning or Markov models on data transformations from users' interactions with channel artifacts, including products and non-product content [43–45]. We extend these methods with cohort behavior, current context, and prescriptive action recommendations. The implementation of this framework is intentionally generalizable, scalable, and operationalizable. These requirements influence the choice of methods we explore.

We summarize our approach to experimenting with trade-offs for user intent prediction and the subsequent corrective prescription as follows:

- Learn the embeddings from the data using long short-term memory (LSTM) [46], an artificial recurrent neural network-related architecture for learning user intent [41]. The embeddings are created by mapping the discrete categorical variables in the listed inputs to vectors of continuous numbers (sequences):
  - aggregated cohort behavior features from the persona generator;
  - user context (context-sensitive variables);
  - user interaction.
- Combine embedding-methods based on linear transformations and concatenation have produced accurate meta-embeddings [47].
- Explore fine-tuning pre-trained BERT (Bidirectional Encoder Representations from Transformers) models with the dataset used for the experiments [48].
- Develop paired cohort-directed prescriptive actions for intent prediction instances that are different from a desired positive outcome.
- Propose testing, validation, and end-to-end architecture for development, deployment, and monitoring.

The user intent detection subsystem models behaviors based on the assumption that the user's interest lasts for some time while browsing the sales channel. Interactions with products indicate the user's interest, and in a single step, the user's click behavior is driven by the user's current interest. The data is modeled as a sequence of click behavior time steps well handled by a LSTM-based neural network. LSTM is adopted for sequence modeling because it has a loop-back mechanism in a forward pass of samples. Through this, it utilizes context from previous predictions for predictions in a new sample. The cell state provides some memory to the LSTM network so it can 'remember' the past. The network has sigmoid activation functions called the input gate, forget gate, and output gate. Figure 6 [49] shows a representation of the LSTM network. The gates, weights, biases, cell states, and final output are formulated as follows:

$$\text{input gate} \;\rightarrow\; i_t \;=\; \sigma(w_i[h_{t-1}, x_t] + b_i)$$
$$\text{forget gate} \;\rightarrow\; f_t \;=\; \sigma\Big(w_f[h_{t-1}, x_t] + b_f\Big)$$
$$\text{output gate} \;\rightarrow\; o_t \;=\; \sigma(w_o[h_{t-1}, x_t] + b_o)$$
$$\text{cell state at timestamp(t)} \;\rightarrow\; \tilde{c}_t \;=\; \tanh(w_c[h_{t-1}, x_t] + b_c)$$
$$\text{candidate for cell state at timestamp(t)} \;\rightarrow\; c_t \;=\; f_t * c_{t-1} + i_t * \tilde{c}_t$$
$$\text{final output} \;\rightarrow\; h_t = o_t * \tanh(c_t)$$

where σ represents the sigmoid function, $w_x$ is the weight of respective gate(x) neurons, $h_{t-1}$ is the output of the previous timestamp, $x_t$ is the input at the current timestamp, and $b_x$ represents the biases for the respective gates(x).

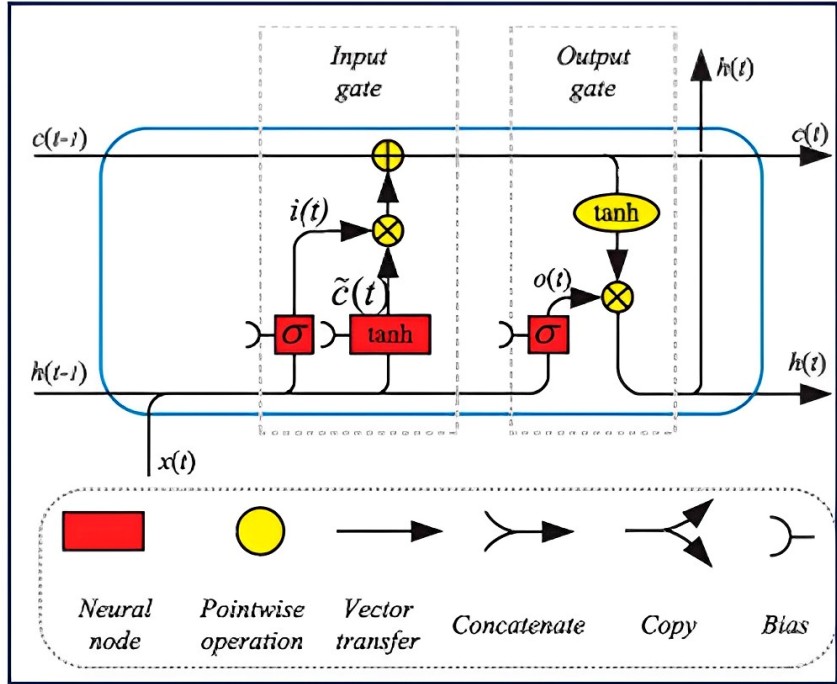

**Figure 6.** Original LSTM architecture.

Data-centric experiments on the implementation of LSTM are extended herein using aggregated cohort metrics and context feature vectors to obtain an improved and generalized performance from the models. Trade-offs and anticipated user experience outcomes are expanded upon in subsequent sections.

*2.5. Cohort-Directed Prescription*

This part of the system generates prescriptive signals for in-session alternative user actions. The vector representation of the prescribed alternative actions is pre-determined before the session and saved in a low-latency database for fast retrieval in the session. These prescribed alternative actions are presented as a sequence of vectors representing the paths typical of members of the same cohort as the user, whose actions on the sales channel led to desired outcomes such as conversion or increased engagement. Figure 7 depicts the implementation of paired cohort-directed prescriptive actions. The vector space is the vector mapping of a set of (m) actions on (n) objects with a dimension (m×n). Typical actions in e-commerce channels include search, view, click, add-to-cart, purchase, addition to wish list, etc. Objects are a combination of products and non-product contents such as webpage types, product category listings, installation guides, and banners.

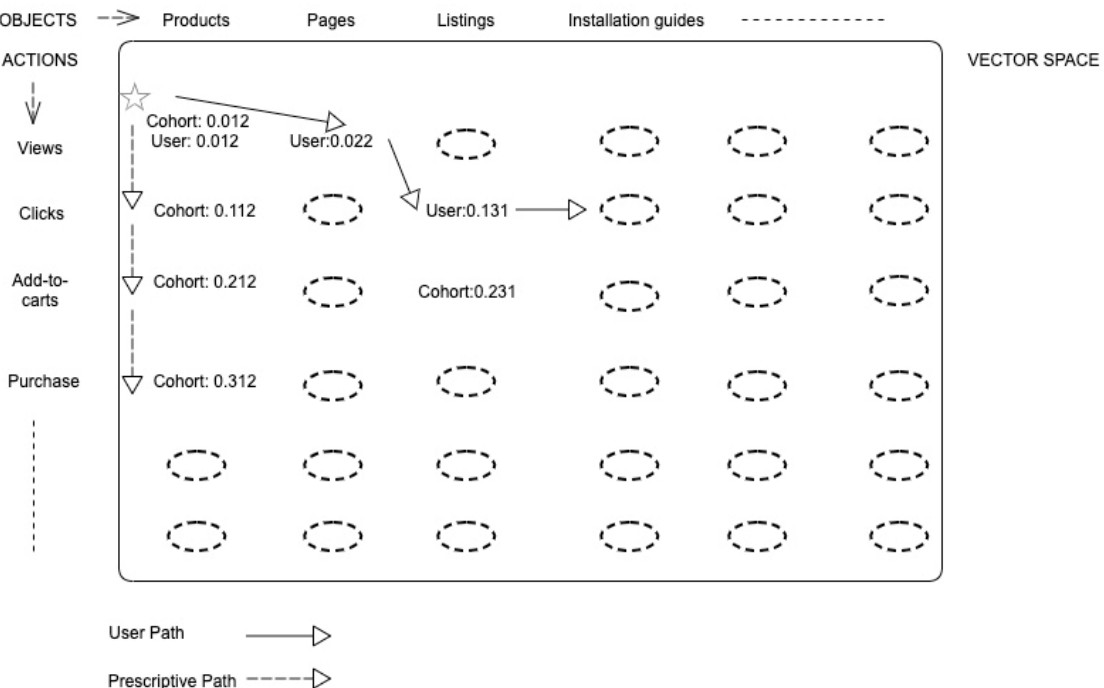

**Figure 7.** Cohort-directed prescriptive actions.

In the illustration, 0.012 is a product view and the symbolic start of the shopping journey of a user that belongs to a cohort whose members that made the most purchases of that product typically follow the path {0.012, 0.112, 0.212, 0.312}. The user's predicted path is the sequence {0.012, 0.022, 0.131, . . .}. Every step on the customer journey is mapped to the vector space. A sequence pattern match will select the cohort sequence {0.012, 0.112, 0.212, 0.312} as a close prescribed match having the same start, and recommend 0.112 (click on the product) as the next best action for the user.

*2.6. Experiments*

The experiment's environment is set up using Python, TensorFlow, and Keras Functional API on the Apple M1 chip of an 8-core CPU with four performance cores and four efficiency cores, an 8-core GPU, and a 16-core Neural Engine. The subset of the data used initially for the deep learning experiment is the first month's users' shopping activities within the first three months: a total of 22.5 million timestamped events. This leaves data from 5 subsequent months of user activities for further experiments and model validation. The dataset is split into 50% training data, 25% test data, and 25% validation data after reshaping to 3D tensors, of which each user sample is a sequence of feature vectors with a length of the number of time steps. The clickstream dataset is unbalanced, with fewer purchases/conversions classes than other event types. By ground-truthing before applying deep learning, we take a naïve, non-machine learning approach to predict that events will be the same as the event of the previous time step. We evaluate this approach using the mean absolute error (MAE) and obtained validation and a test MAE of 0.77 that will serve as a baseline for the deep learning approach.

Combining neural network architecture and training algorithm parameters, especially loss functions, is critical to avoiding subpar model performance and drawing incorrect conclusions from experiments. The neural network error landscape is the subject of ongoing debates and research theories [50]. The loss function computes a score of how far the output is from that expected. The squared-error loss function and cross-entropy loss function are two common examples of loss functions that we considered for suitability. Cross-entropy is often the default loss function for multi-class classification problems. Ideally, for our use case, the complexity of the user intent to be predicted is the cross-multiplication of several

activities on the number of products and services on the shopping channel(s). Modeling this as a multi-class classification problem will be computationally expensive and may be impractical for industry applications. The model is instead structured as a sequence of click behavior time steps. The mean squared error (MSE) is the preferred loss function under the inference framework of maximum likelihood when the target variable has a Gaussian distribution. The experimental results in Section 3 highlight its performance during our implementation of the LSTM network.

Apart from the loss function and the number, size, and types of layers in the network, the optimization procedure and activation function are other parameters under consideration for the application of this experiment. The optimizer lowers the loss score by implementing a backpropagation algorithm to adjust the value of network weights in a direction that decreases the loss. Optimizers are often assigned to two families: gradient descent optimizers, such as stochastic gradient descent and batch gradient descent, and the adaptive variants of SGD, such as Adagrad, Adadelta, RMSprop, and Adam [51,52]. It is difficult to tune the learning rate of gradient descent optimizers, and they have a high risk of getting stuck in suboptimal local minima. For these reasons, our preference was for adaptive optimizers. We present experimental results using the RMSprop and Adam optimizers in Section 3.

Word2Vec embeddings (vector space word representations) were used to obtain a vectorized representation of the categorical features. Word2Vec groups together the vectors of similar words and can make robust estimates of the meaning of the words based on their occurrences in a large enough corpus of text [53]. The dataset was sampled in batches of pre-defined sequence lengths while running the deep learning experiments. This process enables flexibility and a trade-off between the number of sequence batches and computation time for each epoch. The different parameters in the experiments and the resulting trade-offs in performance and computation time are discussed further in Section 3 of this paper. We performed experiments with an early stopping callback to interrupt training when validation loss had stopped improving and to save the best model in each iteration. This process helps prevent overfitting during training. The callback object is passed to the model in the call to fit( ) and has access to the model's state and its performance throughout the iteration.

Data-centric iterations were carried out on the model, extending the features with cohort metrics and context variables from the persona generator and user context subsystems to validate that these extensions improve the user intent prediction. We also used pre-trained BERT model APIs to run end-to-end tokenization of the categorical features to compare with the model's performance using Word2Vec embeddings. The performance metrics from the data-centric iterations in the user intent prediction subsystem are highlighted in the next section.

### 3. Results

The dataset was fed into the LSTM model during the first iteration using a sampling rate of 10, with a batch size of 1280 and a sequence length of 360. The baseline LSTM had one hidden layer with 16 neurons, one dense output layer, RMSprop optimizer, and mean squared error as the loss function. The performance was measured using mean absolute error. The categorical features were handled via a categorical encoder assigning different numeric values for each category. With these parameters, the baseline LSTM was run in 10 epochs, with the 8777 steps in each epoch taking about 390 ms/step. Model overfitting was indicated by the training loss, which stayed at 0.0202 from the third epoch onward. This baseline model delivered the best mean absolute error (MAE) of 0.0495 on the training dataset and 0.05 on the test dataset. Increasing the neurons in the hidden layer to 32 on the second iteration did not significantly impact performance. Instead, it increased computation time for the same level of performance. The parameters and computation times of these two iterations of the baseline model are highlighted in Table 3.

**Table 3.** Increasing the number of neurons in the hidden layer of the baseline LSTM model.

| Iteration | Steps/Epoch | Average Time/Step | No. of Neurons | Optimizer | Loss Function | Loss | MAE | Test MAE |
|---|---|---|---|---|---|---|---|---|
| 1 | 8777 | 390 ms | 16 | RMSprop | MSE | 0.0194 | 0.0495 | 0.050 |
| 2 | 8777 | 10,262 s | 32 | RMSprop | MSE | 0.0202 | 0.0479 | 0.055 |

We adopted the classic technique of fighting overfitting with dropout in both iterations. The same dropout mask was applied at every time step, allowing the network to propagate its learning error over time. As seen in the plots of training MAE and validation MAE in Figure 8, the model starts to become overfitted with the beginning of the third epoch in both iterations, indicating that the subsystem would benefit from a mechanism to stop model training once it overfits.

Iteration 1

Iteration 2

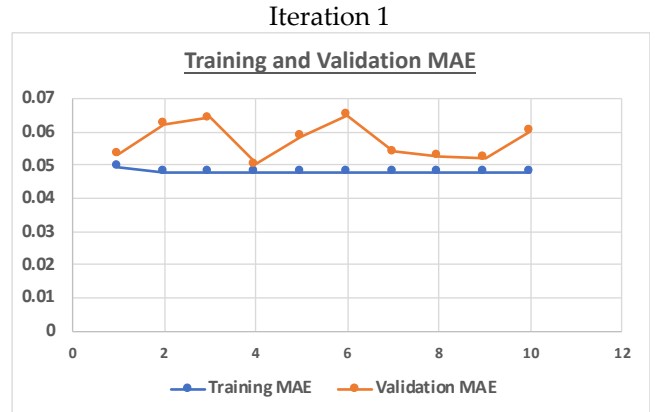
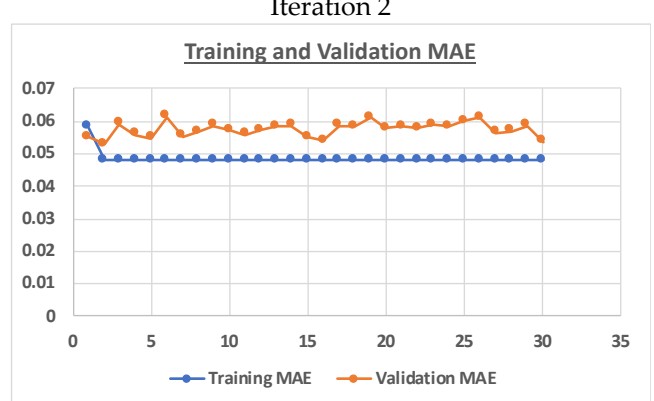

**Figure 8.** Plots of training and validation mean absolute error.

For the third iteration, Word2Vec embeddings were used to create a vectorized representation of the categorical features; the optimizer was changed to Adam for computational efficiency, the accuracy was included in the measured metrics, the sampling rate was increased to 12 and the sequence length to 720, but the batch size was reduced to 256. The neurons in the hidden layer were reduced to 4 for the fourth iteration, cohort metrics and early stopping were included, and the batch size was set back to 1280. Details of the configurations of these subsequent iterations are highlighted in Table 4. As seen in the metadata, adding only cohort metrics slightly reduces the model's generalizability with a marginally higher validation MAE.

**Table 4.** Including cohort metrics in the features.

| Iteration | Steps/Epoch | Average Time/Step | No. of Neurons | Optimizer | Loss Function | Loss | MAE | Accuracy | Test MAE |
|---|---|---|---|---|---|---|---|---|---|
| 3 | 43,864 | 265 ms | 16 | Adam | MSE | 0.0194 | 0.0419 | 0.9688 | 0.04 |
| 4 (+Cohort Metrics) | 8773 | 377 ms | 4 | Adam | MSE | 0.0194 | 0.0465 | 0.9688 | 0.05 |

The subsequent iterations (5, 6, and 7) include context variables, derived context variables, and then a combination of both cohort metrics and context variables. Figure 9 shows the structure of iteration 7 with combined cohort metrics and context variables.

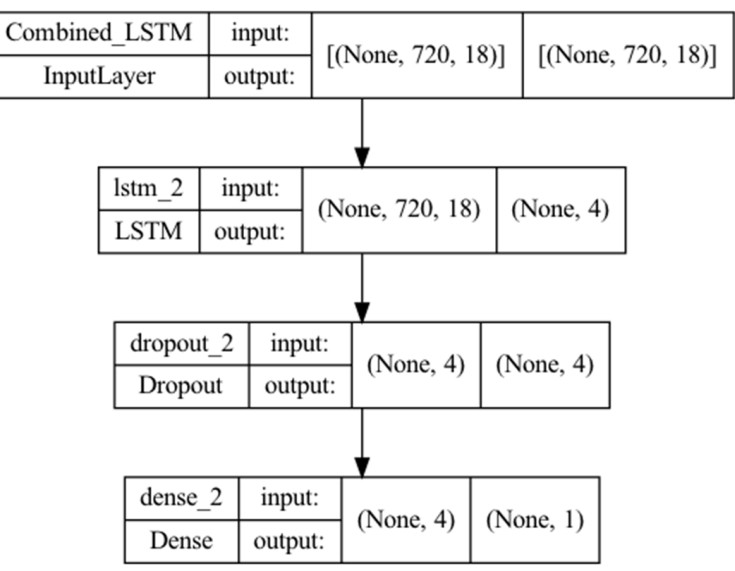

**Figure 9.** Structure of iteration 7—combined cohort metrics and context variables.

In iteration 8, a pre-trained BERT model was used to tokenize the categorical features, and then these contextual embeddings were used as inputs for the LSTM model. Table 5 compares the results from iterations 3 to 8.

**Table 5.** Comparing multiple data-centric iterations.

| Iteration | Steps/ Epoch | Average Time/Step | No. of Neurons | Optimizer | Loss Function | Loss | MAE | Accuracy | Test MAE |
|---|---|---|---|---|---|---|---|---|---|
| 3 | 43,864 | 265 ms | 16 | Adam | MSE | 0.019 | 0.042 | 0.9688 | 0.04 |
| 4 | 8773 | 377 ms | 4 | Adam | MSE | 0.019 | 0.047 | 0.9688 | 0.05 |
| 5 | 8773 | 394 ms | 4 | Adam | MSE | 0.019 | 0.048 | 0.9688 | 0.05 |
| 6 | 8773 | 413 ms | 4 | Adam | MSE | 0.019 | 0.049 | 0.9688 | 0.05 |
| 7 | 8773 | 436 ms | 4 | Adam | MSE | 0.019 | 0.049 | 0.9688 | 0.05 |
| 8 | 1952 | 431 ms | 4 | Adam | MSE | 0.015 | 0.043 | 0.9731 | 0.04 |
| 4 | Cohort Metrics only | | 7 | Combined Cohort Metrics and Context Variables | | | | | |
| 5 | Context Variables only | | 8 | BERT Pre-trained Model | | | | | |
| 6 | With Derived Context Variables | | | | | | | | |

Adding context variables and derived context variables appear not to impact the model's performance significantly. Combining cohort metrics and context variables did not have a significant impact either. The validation MAE increases slightly for iterations 4 to 7. Iteration 8 of the experiment, which leverages a pre-trained BERT model, gave the best performance overall with a 0.44% increase in accuracy and 0.01 or 20% decrease in the test dataset MAE compared to other iterations (apart from iteration 3, which had matching metrics for the test dataset MAE). The metrics indicate a lower MAE on the validation dataset than for iterations 4 to 7. Interestingly, the lowest MAE was recorded on the validation and test datasets for iteration 3, which did not include cohort metrics or context variables.

## 4. Discussion

All of the experimental iterations had reasonable predictive accuracy and indicated the framework's usability for implementing personalization. Arguably, cohort metrics and context variables did not impact the model performance as implemented with this dataset because the cohort metrics and context variables were also computed from features already in the model. In practice, and as indicated in Section 2.3, which covered the context-awareness computing module, user searches are a rich source of the user's current context. Including the contextual embeddings of user searches positively impacts user intent prediction, as validated in actual practice, while adopting components of this framework to implement personalization to benefit retail customers. However, due to certain data privacy and policy restrictions on using the retail company's customer data for research, we were limited to implementing this framework with publicly available data so that the results could be shared publicly and, if desired, replicated.

The end-to-end implementation of personalization, as demonstrated with the proposed systems engineering framework, fulfills the hitherto unfulfilled promise of many solution vendors providing e-commerce. Vendors often offer consulting services that span several months spent on implementing some semblance of personalization. However, apart from the cost and time, the solutions leverage proprietary technologies and often only satisfy parts of the end-to-end delivery. Similarly, several studies have proposed solutions to context-aware personalized recommendations. However, their focus has been on aspects of the theory and they are limited primarily to recommender systems. For instance, the paper in [3] measures how much contextual information matters in building models for personalization applications. In contrast, the focus in papers [2,4,5] is limited to recommender systems with applications for in-car music [5] and tourism [2].

This systems engineering framework covers the end-to-end practical implementation of personalization that leverages privacy-compliant customer profiling methods, adopts current context, preempts purchase intent, and influences customer in-session action with cohort-directed prescriptions for the next best action. Moreover, the framework is loosely coupled, so its different subsystems can be implemented independently in a microservice style to serve other purposes. For instance, the persona generator develops different groupings of customers that are alike, and it can be leveraged for targeted marketing of customer cohorts with relevant ads, thereby enabling personalized marketing. Similarly, the module's logic that generates context-aware signals is adaptable to different applications, including those beyond retail. The framework's components are open-source so that industry practitioners can adopt it. In practice, parts of the framework have been implemented to scale for more than 2 million retail products and 8 million plus daily users on an open-source distributed computing database.

The research findings from the data-centric iterations on the implementation of intent detection highlight the value of modeling users' online actions as sequences. Its best result of 97.3% accuracy indicates that the method developed is effective. This is an improvement over initial results obtained during preliminary explorations that obtained accuracy scores of up to 79% using decision tree-based ensembles to classify shopper intent. It also compares

favorably with the range of scores in the experiments in [54] that apply different classifier models to early purchase prediction.

The implementation of the cohort-directed prescription for alternative user actions while in a shopping session scales very well across products and users and is functional for creating users' next best actions at every point in their shopping journey. It adjusts at each step of the shopping journey, with awareness of the last step to iteratively compute the next step to be prompted, leading to improved user experience, engagement, and conversion.

The systems engineering framework has a level of abstraction that enables portability and scalability, improving many systems that leverage artificial intelligence and are specific to defined problems. The framework's components are loosely coupled subsystems that can be swapped with improved implementations as technology evolves. This has more significant ramifications for adopting this framework in implementing personalization in the industry, as component microservices can be swapped out with more effective and efficient components in the future.

As indicated previously, data privacy and policy restrictions prevented the use of a retail company's customer data to demonstrate this framework. Another limitation of this study is that it did not experiment with implementing intent prediction with high-order Markov chain models, which are also apt for modeling sequences.

The modular design of the framework, using open-source technologies, makes it flexible enough to swap the tech stacks used for implementation as technologies mature and improve. Implementation of the intent detection subsystem using Markov chain models would be an interesting direction for future research. It will be worthwhile to examine whether it can be implemented to scale in production and yield better predictive values. Similarly, to further this research, variational deep embedding (VaDE) [55], which leverages unsupervised generative clustering within the framework of a variational auto-encoder (VAE), may be considered for the persona generator subsystem since it allows more general mixture models than GMM to be plugged in.

## 5. Conclusions

In this study, we established a framework for the systems engineering of an intelligent system that leverages user behavior and current session context to predict the user's intent and prescribe cohort-directed next-best actions at every point in the user's journey. We investigated an ensemble of algorithms structured to serve as microservice-styled subsystems of the framework, which comprised the persona generator, context awareness, and intent prediction subsystems. Customer behavior is modeled as sequences well handled by a data-centric extension of the long short-term memory artificial recurrent neural network, achieving a predictive accuracy of 97.3%.

While this study provides valuable insights into the subject, it is important to acknowledge its limitations. The dataset used is not fully representative of the industry at large. A typical industry dataset will have more variations of possible shopper actions and contexts, thereby creating more complexity for the system to handle. Similarly, in instances with underwhelming information on customer behavior, products, and preferences, the system will not have enough input to be 'intelligent'. It is important to also note that model complexity and scale will make the proposed solution not feasible for industry practitioners with limited computation capacity.

The system facilitates positive and productive shopping experiences for the users by 'knowing' what the users want and helping them to find what they are looking for faster, mimicking what is obtainable from the attention of a knowledgeable in-store shop assistant. The user receives more relevant content, improved search ranking, and personalized product recommendations. As a result, the business achieves improved customer engagement, more sales, and greater customer loyalty. The solution models complex user behavior using cutting-edge machine learning but follows fundamental systems engineering principles using scalable, replicable, and accessible open-source technology. This study is valuable for

the experiments' research findings and the outcome—an end-to-end practical and scalable implementation of personalization that practitioners can leverage in the retail industry.

**Author Contributions:** Conceptualization, O.O. and S.S.; methodology, O.O. and S.S.; software, O.O.; validation, O.O. and S.S.; formal analysis, O.O.; investigation, O.O.; resources, O.O.; data curation, O.O.; writing—original draft preparation, O.O.; writing—review and editing, S.S.; visualization, O.O.; supervision, S.S.; project administration, S.S. All authors have read and agreed to the published version of the manuscript.

**Funding:** This research received no external funding.

**Data Availability Statement:** Publicly available datasets were analyzed in this study. This data can be found here: https://www.kaggle.com/datasets/mkechinov/ecommerce-behavior-data-from-multi-category-store (accessed on 20 July 2023).

**Acknowledgments:** We wish to show our appreciation to Michael Savoy of Lowe's Home Improvement data science, for his review, especially on the feasibility of the proposed framework.

**Conflicts of Interest:** The authors declare no conflict of interest.

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
