# Peer review of "Context-Aware Personalization: A Systems Engineering Framework"

_information, doi:10.3390/info14110608_

Round 1
Reviewer 1 Report
Comments and Suggestions for Authors
The authors proposes a framework based on Systems Engineering that can be practical and scalable for personalization to leverage privacy-compliant customer profiling methods for best next content.
The introduction part can be improved in terms of listing proposed contributions in an enumerated manner that will enable users to focus on that part.
For example, the sentence 'The objective is to enable a system that fits loosely coupled microservice-based e-commerce platforms and produces signals that activate products and non-product responses personalized to individual users in the current session' is too complex to make sense. Please break this and eloborate it What do you want to say as a reader who doesn't need to be an expert in your domain to understand it?
Line 214: Are there any experimental results for the selected number's feasibility?
Line 217: Why?
Who developed this API environment? Based on what the API response back context variables?
330. Logistic regression classifier predicting purchase: what are the details of the logistic regression model?
465 Word2Vec: Please give more detail.
In summary, the framework and combining all these steps for personalised content are interesting. However, more technical detail on the models and their implementation, including their parameters, needs to be provided.
Also, do you have an estimate of the total time spent on providing the next content after running the framework (from the beginning to the end steps)?
Comments on the Quality of English LanguageThe quality of the English is acceptable.
Author Response
Dear Reviewer,
Thank you for the rigorous review and the helpful feedback. The responses are attached.

Reviewer 2 Report
Comments and Suggestions for Authors
The authors have proposed a framework for developing e-commerce systems based on context-aware personalization. The framework allows for modeling user behavior based on their interactions with sales channels and media. The user behavior-based content is generated using three components: knowledge of the users and their behavior, awareness of users’ current context, and the comprehension of the situation and prediction of the future user's intentions. The first component uses an unsupervised machine learning algorithm to assign users into cohorts and learn cohort behavior. The second component applies some novel e-commerce data interpretation. And finally, the prediction component uses a pre-trained BERT model for tokenizing categorical features, which are then used as input for the LSTM network.
Please address the following issues:
1. The authors have described the proposed framework rather quite well, although more detailed diagrams showing its architecture should be provided. The diagram presented in Fig. 5 is very general.
2. Can the framework be downloaded, used in practice and experimentally verified by other researchers?
3. The experimental part of the paper is not very extensive. More experiments, probably using more realistic data, should be carried out.
4. Were some other machine learning models used for prediction? Why were the particular models selected?
5. Can the proposed framework be used in other areas than e-commerce? If yes, please provide some examples or usage scenarios.
Please improve the English language, especially grammar and style.
Author Response
Dear Reviewer,
Thank you for the rigorous review and the very helpful feedback. The article has been reviewed as recommended, and the responses to comments are attached here.

Reviewer 3 Report
Comments and Suggestions for Authors
Comments:
By analyzing the manuscript information-2668258 we can asseverate that this study proposes a framework for a system engineering approach to context-aware personalization context-aware personalization applied to e-commerce, understanding and modeling user behavior from their interactions with sales channels and media. The structure is practical and based on systems engineering principles. After reading the text, we can make the comments described below:
1. The work is well structured, the research questions and objectives are present in the introduction. The methodology allows other researchers to reproduce it. The discussion and conclusion are appropriate.
2. In formal terms, the equations should be centralized and properly numbered. Proofread the entire article;
3. I suggest breaking down figure 9 into a figure and table. The source of this figure is missing. Please indicate the source as well.
4. The authors have registered the references in the APA method in the references section, I suggest that you revise this entire section and format it according to the instructions for the authors.
5. Indicate whether the model script was developed and which programming language was used.
6. Indicate whether the proposed model has been patented.
7. I suggest inserting a list of all the abbreviations used in the text at the end of the paper.
Good review
Author Response

(The authors gave the same response as above.)
